# Comparison of the effects of exergaming and balance training on dynamic postural stability during jump-landing in recreational athletes with chronic ankle instability

**Sadaf Sepasgozar Sarkhosh, Roya Khanmohammadi●\*, Zeinab Shiravi**

Department of Physical Therapy, Tehran University of Medical Sciences, Tehran, Iran

\* rkhanmohammadi@sina.tums.ac.ir

**Editor:** Žiga Kozinc, Faculty of Health Sciences, University of Primorska, SLOVENIA

**Data Availability Statement:** The data supporting the findings of this study can be found in the Supporting Information files of the manuscript.

## Abstract

The primary inquiry of this study was to determine if exergaming is more effective than balance training in improving dynamic postural control during jump-landing movements among athletes with chronic ankle instability (CAI). Additionally, the study aimed to compare the effectiveness of these interventions on clinical and psychological outcomes. This study was a randomized, single-blinded, controlled trial in which participants were assigned to either an exergaming group or a balance training group. Outcome measures were assessed before, after, and one month following the intervention. Primary outcomes included the stability index (SI) and time to stabilization (TTS) in the anteroposterior (AP), mediolateral (ML), and vertical (V) directions, the dynamic postural stability index (DPSI), and the resultant vector time to stabilization (RVTTS). Secondary outcomes included performance, fear of movement, and perceived ankle instability, measured using the side-hop test, the Tampa Scale for Kinesiophobia (TSK), and the Cumberland Ankle Instability Tool (CAIT), respectively. Results indicated a significant decrease in ML SI in both groups one month after treatment compared to before and after treatment (P = 0.013 and P<0.001, respectively). Additionally, one-way ANCOVA revealed a significant difference between the groups post-treatment (F (1,31) = 6.011, P = 0.020, η2 = 0.162) and one month post-treatment (F(1,31) = 4.889, P = 0.035, η2 = 0.136), with ML SI being significantly lower in the exergaming group than the balance training group at both time points. In both group, the DPSI also decreased significantly one month post-treatment compared to before and after treatment (P = 0.040 and P = 0.018, respectively). Both groups showed improvements in performance, severity of perceived ankle instability, and fear of movement. Thus, the study concluded that both exergaming and balance training effectively improved postural control during jump-landing, with exergaming showing superior performance in the ML direction both after and one month post-treatment. In terms of clinical and psychological outcomes, both interventions were effective, with neither showing superiority over the other.

**Funding:** This project was supported by a grant from Tehran University of Medical Sciences (Grant No. 1401-2-103-66959). The funders had no role in study design, data collection and analysis, decision to publish, or preparation of the manuscript.

**Competing interests:** The authors have declared that no competing interests exist.

## 1. Introduction

Ankle sprains are among the most prevalent musculoskeletal injuries, and up to 70% of individuals who experience one may develop residual physical disabilities, potentially leading to chronic ankle instability (CAI) [1]. CAI is characterized by recurrent ankle sprains and frequent sensations of the joint giving way, leading to prolonged absences from sports activities and financial losses [2].

Several studies, in conjunction with Freeman's research in the mid-1960s, have indicated a strong correlation between CAI and insufficient postural control, a crucial element in sports activities [3–5]. Therefore, researchers have focused on enhancing postural control, as it is known to be a contributing factor in recurrent sprains and increased risk of re-injury [5].

From a therapeutic perspective, various treatments have been suggested to enhance postural control in individuals with CAI. One treatment that has garnered attention in recent years is exergaming. The term "exergame" is a combination of "exercise" and "game," referring to the use of computer games to increase physical activity levels [6]. Exergaming is an innovative technology that is currently being developed in the field of rehabilitation [6].

While traditional balance training have been demonstrated efficacy in enhancing postural control [7], they often suffer from the drawbacks of being somewhat monotonous, failing to capture participants' interest, and sometimes leading individuals to discontinue treatment prematurely before achieving desired results [8–10]. Therefore, an attractive and entertaining treatment is required. Exergaming is an attractive option. Moreover, in traditional balance training, individuals may not receive prompt feedback, which can discourage active involvement. Conversely, in exergaming, individuals receive continuous visual and auditory feedback, thereby increasing the level of engagement [11]. In addition, instantaneous feedback enables users to concentrate more on their movements. Within gaming environments, feedback is given based on player's performance and outcomes, potentially enhancing learning of motor tasks [12].Moreover, exergaming has the potential to simultaneously engage various cognitive and motor resources [13].The engagement of cognitive resources is a significant advantage of this rehabilitation program. Studies have indicated that individuals with CAI experience impaired information processing, with cognitive demand identified as a factor contributing to decreased performance and recurrent injury [14–16]. Consequently, it seems that enhancing motor skills while simultaneously reducing reliance on conscious information processing through motor-cognitivetraining could enhance the processing capacity of the central nervous system (CNS).Enhancing processing capacity enables the CNS to effectively process a greater amount of intrinsic and extrinsic information, thereby possibly improving ankle stability and preventing re-injury [15]. Another benefit of exergaming is that it requires users to continually adjust their weight in various directions, at different speeds and intensities while keeping their center of gravity (COG) on the base of support. These controlled movements closely mimic the trunk, hip, and ankle strategies [17].

Few studies have investigated the effectiveness of exergaming in individuals with CAI on postural control. These studies suggested using exergaming in rehabilitation programs [18, 19]. However, in these studies, postural stability was assessed through simple tasks where participants stood and attempted to maintain their COG on a base of support. Yet, for athletes, maintaining dynamic postural stability during more demanding tasks is of greater importance. Dynamic postural stability involves maintaining balance while transitioning from dynamic to static positions [20]. Simple tests do not adequately challenge the neuromuscular system and may overlook inadequate postural stability due to their straightforward nature [20]. In contrast, activities like jump-landing provide a more appropriate challenge to the neuromuscular

system [21]. In addition, movements like jump-landing closely resemble sports activities, during which the majority of ankle injuries tend to occur [22].

Studies have shown that the kinematics and kinetics of jump-landing is altered in individuals with CAI [23–25]. The stability index (SI) and the time to stabilization (TTS) are two appropriate metrics for assessing dynamic postural stability in jump-landing. According to a 2019 meta-analysis, individuals with CAI exhibit higher scores on the dynamic postural stability index and require more time to achieve stability compared to healthy individuals. This study reported a medium to large effect size (0.45–0.74) for these metrics [25].Additionally, these differences in jump-landing strategies, relative to those observed in healthy individuals, are associated with an increased risk of recurrent injuries [26]. Therefore, improving these strategies and ensuring adequate stability during jump landings can be a crucial component of the rehabilitation program.

Despite the importance of dynamic postural control during jump-landing and the growing utilization of exergaming in sports communities as part of rehabilitation regimens, there is no research on the effectiveness of this intervention on this issue in athletes with CAI. Consequently, the primary inquiry of this study was whether, among athletes dealing with CAI, exergaming is superior to traditional balance training in improving dynamic postural control during jump-landing movements, which are challenging and functionally important for this population. In addition, another aim of the study was to compare the effectiveness of these groups on clinical and psychological outcome measures.

## 2. Method

### 2.1. Study design

This study was conducted as a randomized, single-blinded, controlled trial. Participants were assigned to one of two groups: one receiving exergaming and the other performing balance training. Outcome measures were assessed before the intervention, after, and one month following the treatment. Post-tests were conducted within 24 to 48 hours after the completion of the treatment. Both groups underwent individual treatment sessions in a laboratory environment.

The primary outcome measures included the SI and TTS in the anteroposterior (AP), mediolateral (ML), and vertical (V) directions, along with the dynamic postural stability index (DPSI) and the resultant vector time to stabilization (RVTTS) during jump landing. The SI and TTS were selected for their strong reliability as established metrics for evaluating dynamic balance during jump landings. These parameters can distinguish individuals with CAI from those who are healthy, as research show that individuals with CAI typically achieve higher scores on the SI and TTS than their healthy counterparts [25]. Furthermore, these parameters are associated with an increased risk of recurrent injuries [26].

These measures enable the evaluation of neuromuscular control, which is crucial in minimizing the risk of re-injury and improving functional outcomes in sports. Thus, improving these parameters and ensuring sufficient stability during jump landings is a vital aspect of the rehabilitation program.

Secondary outcomes included performance metrics, fear of movement, and the severity of perceived ankle instability, assessed using the side-hop test, the Tampa Scale for Kinesiophobia (TSK), and the Cumberland Ankle Instability Tool (CAIT), respectively. These secondary outcomes were selected to provide a more holistic evaluation of the participants' functional and psychological well-being.

The side-hop test measures agility and functional performance of the ankle during dynamic activities, while the TSK evaluates fear of movement or re-injury—an important factor that

can significantly impact rehabilitation and athletic performance. The CAIT assesses participants' subjective perceptions of ankle instability, which is essential for understanding their overall ankle function. Incorporating these secondary measures enhances the primary dynamic postural stability outcomes by considering both functional performance and psychological factors that play a crucial role in the rehabilitation process and long-term recovery for athletes with CAI.

The subjects were recruited between December 24, 2023, and May 25, 2024. This study was registered as a clinical trial on the Iranian Registry of Clinical Trials (IRCT20230124057209N1).

## 2.2. Participants

Athletes with CAIwere recruited from the university community, sports clubs, and federations using posters and social media advertisements (Table 1). A flow diagram based on the CONSORT statement illustrates the participants' progress from enrollment to analysis (Fig 1). All participants provided written informed consent, and the study received approval from the Tehran University of Medical Sciences Ethics Committee (IR.TUMS.FNM.REC.1401.146).

### 2.2.1. Inclusion criteria.

1. Age between 18 and 40 years.

2. A self-reported history of at least one significant acute ankle sprain that:

   ○ Occurred more than 12 months prior to study enrollment.

   ○ Was associated with inflammatory symptoms, such as pain and swelling.

   ○ Resulted in at least one day of interrupted physical activity [2].

3. The most recent ankle sprain occurred more than 3 months before study enrollment [2].

**Table 1. The demographical and clinical characteristics of subjects at baseline.**

| Characteristics | Intervention (N = 17) | | Control (N = 17) | | P |
|---|---|---|---|---|---|
| | Mean / Median | SD / Q1–Q3 | Mean / Median | SD / Q1–Q3 | |
| Age (years)* | 22.00 | 21.00–24.00 | 24.00 | 21.00–36.00 | 0.114 |
| Height (cm)″ | 176.53 | 7.24 | 174.00 | 9.61 | 0.393 |
| Weight (kg)″ | 69.59 | 9.79 | 70.18 | 11.18 | 0.871 |
| CAIT (0–30)″ | 16.28 | 4.34 | 15.33 | 5.30 | 0.570 |
| Time since first sprain (months)″ | 16.12 | 3.59 | 16.71 | 4.04 | 0.657 |
| Time since last sprain (months)* | 3.00 | 3.00–5.50 | 4.00 | 3.00–12.00 | 0.306 |
| Number of giving ways″ | 8.18 | 4.69 | 6.94 | 4.16 | 0.423 |
| Physical activity (hours/week)″ | 10.79 | 4.98 | 9.00 | 5.02 | 0.303 |
| Female (N (%))✦ | 3 (18) | | 8 (47) | | 0.067 |
| Right affected side (N (%))✦ | 9 (53) | | 9 (53) | | 1.000 |
| Right dominant side (N (%))✦ | 14 (82) | | 15 (88) | | 0.628 |

CAIT = Cumberland Ankle Instability Tool.

Q1 and Q3 represent the first and third quartiles of a parameter.

✦Chi-square test was used.

″ Independent t-test was used.

* Mann-Whitney U test was applied.

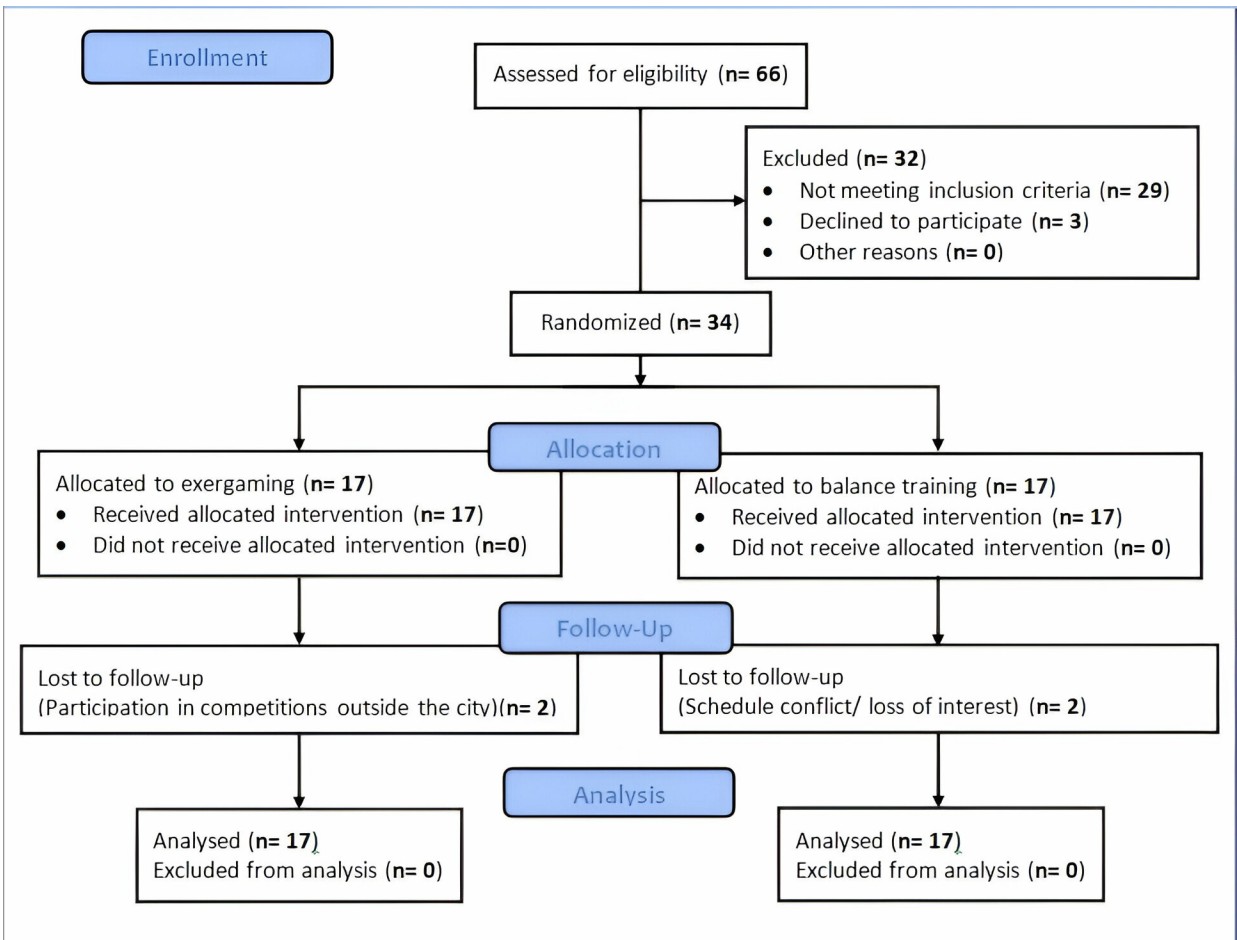

**Fig 1. CONSORT flowchart.**

4. A history of at least two episodes of "giving way," recurrent ankle sprains, or "feelings of ankle joint instability" within the last 6 months:

   ○ "Giving way" refers to uncontrolled and unpredictable episodes of excessive inversion of the rear foot during activities, which do not result in an acute sprain.

   ○ "Recurrent sprains" are defined as two or more sprains to the same ankle.

   ○ "Feelings of instability" refer to the perception of instability during daily activities or sports, often associated with the fear of a new ankle sprain [2].

5. A Cumberland Ankle Instability Tool (CAIT) score of ≤ 24 (the CAIT is scored from 0–30, with lower scores indicating more instability) [2].

6. Participants must be recreationally active, engaging in sports that involve jumping (such as volleyball, basketball, soccer, or handball) at least three times a week, for a minimum of 30 minutes per session [27].

7. No self-reported history of fractures or surgery in the lower extremity.

8. No known psychological or neurological disorders.

9. No prior experience with video game-based training.

### 2.2.2. Exclusion criteria.

1. If participants missed three or more non-consecutive training sessions, or two consecutive sessions.

2. They unable to perform the required maneuvers.

## 2.3. Sample size

The sample size was estimated from a pilot study, focusing on SI and TTS in the ML direction. An a priori power analysis was conducted using G*Power 3.1.9.2 software to achieve 80% power. The statistical testing method chosen was a repeated measures ANOVA with a within-between interaction design. The calculation used a effect size of 0.054 and 0.061, a power of 0.8, $\alpha = 0.05$, non-sphericity correction ($\epsilon$) = 1, and a correlation among repeated measures of 0.5. The analysis indicated that at least 30 participants were needed to detect a within-between interaction effect in a test design of 2 groups and 3 measurements with the specified parameters. Consequently, 34 participants were recruited to account for a potential 10% dropout rate.

## 2.4. Randomization and blinding

The participants were randomly assigned to either the intervention group (exergaming) or the control group (balance training) with a 1:1 allocation ratio. Block randomization (block size = 4) was performed using a web-based randomization service (www.randomization.com). Random allocation was carried out by a third party not involved in the study. Details of the allocated groups were written on cards and concealed using sequentially numbered, opaque, sealed envelopes. In this way, the allocation sequence was concealed from the main researchers. In this study, the assessor was blinded to group allocation.

## 2.5. Assessments

**2.5.1. The SI and TTS extracted from jump-landing.** A force plate device (Bertec Corporation, Columbus, OH, USA) was utilized with a sampling frequency of 500 Hz to test lateral jump landings. The Bertec force plate is well-known for its exceptional reliability and validity in assessing ground reaction forces (GRF). Numerous studies have shown that it provides accurate and consistent data across different movement patterns, establishing it as a "gold standard" for evaluating balance and center of pressure (COP) measurements [28–30]. To determine the maximum horizontal jump, participants performed lateral jumps to achieve the greatest possible distance. The maximum distance achieved was recorded. For safety, 75% of this maximum distance was used for jumps on the force plate. Participants stood on both feet at a distance of 75% of their maximum horizontal jump from the center of the force plate, facing forward with hands on their hips, and subsequently landed on the affected limb [31]. Participants were instructed to regain stability as quickly as possible following landing.

Before the main test, participants performed several practice jumps. If a participant failed to maintain balance upon landing, if the non-landing limb interfered during the jump or landing, if there was an additional small hop on landing, or if there was excessive movement in the arms, trunk, or non-landing limb causing the test limb to lift off the force plate, the jump was discarded and repeated. Data from three successful jumps, including ground reaction force in the x, y, and z directions, were recorded on the force plate for 10 seconds post-landing. There was a one-minute interval between each repetition [21].

The stability index in the ML and AP directions was calculated by measuring the deviation of the X and Y components of the GRF from the zero point, respectively, using Eqs 1 and 2. The stability index in the vertical direction was determined by measuring the deviation of the Z component of the GRF from the individual's weight (Eq 3). The dynamic postural stability index combines the stability indices from the AP, ML, and vertical directions, reflecting sensitivity to changes in all three directions (Eq 4). These values were normalized based on each individual's weight to allow for comparisons between participants. The average of three repetitions was used as the final data. Calculations were based on the period from the moment of landing (when the vertical component of the GRF exceeded 5% of body weight) to three seconds afterward, as this duration best mimics sports performance [21]. Moreover, to determine body weight, participants stood on the force plate on one limb for 5 seconds, and the average vertical GRF during this time was considered as the individual's weight [21].

$$\mathbf{MLSI} = \sqrt{\left(\frac{\sum (0 - GRFx)^2}{number\ of\ data\ points}\right)} \div BW \qquad \text{Eq1}$$

$$\mathbf{APSI} = \sqrt{\left(\frac{\sum (0 - GRFy)^2}{number\ of\ data\ points}\right)} \div BW \qquad \text{Eq2}$$

$$\mathbf{VSI} = \sqrt{\left(\frac{\sum (BW - GRFz)^2}{number\ of\ data\ points}\right)} \div BW \qquad \text{Eq3}$$

$$\mathbf{DPSI} = \sqrt{\left(\frac{\sum (0 - GRFx)^2 + \sum (0 - GRFy)^2 + \sum (BW - GRFz)^2}{number\ of\ data\ points}\right)} \div BW \qquad \text{Eq4}$$

To calculate the time to stabilization, a time series was created by sequentially averaging the normalized GRFs in the AP, ML, and vertical directions. This method involved adding each successive GRF data point to the previous one and averaging them. This process produced a time series from the sequential averaging of the GRFs. Additionally, the standard deviation and average time series of the normalized GRF in the AP and ML directions during the first 3 seconds after landing were calculated. A threshold was set as "standard deviation ± 0.25 of the mean." Stability was indicated when the time series from sequential averaging fell within this threshold range. The time point at which the time series entered this range was considered the moment of stability, and the interval between this moment and the landing moment was defined as the TTS (Fig 2). For the vertical GRF, the threshold was "±5% of the individual's weight" (Fig 3). The RVTTS was also calculated using Eq 5, and the average of three repetitions was considered the final data for each direction [21].

$$\mathbf{RVTTS} = \sqrt{(MLTTS)^2 + (APTTS)^2} \qquad \text{Eq5}$$

**2.5.2. Side-hop test.** The side-hop test was used to assess performance. In this test, participants are instructed to land laterally on the affected limb at a distance of 30 cm. This process is repeated ten times, each repetition consisting of a back-and-forth movement. Participants were asked to complete the task as quickly as possible and the time was measured using a stopwatch. This test has an intra-class correlation coefficient (ICC) of 0.84 and a standard error of measurement (SEM) of 2.10 [32].

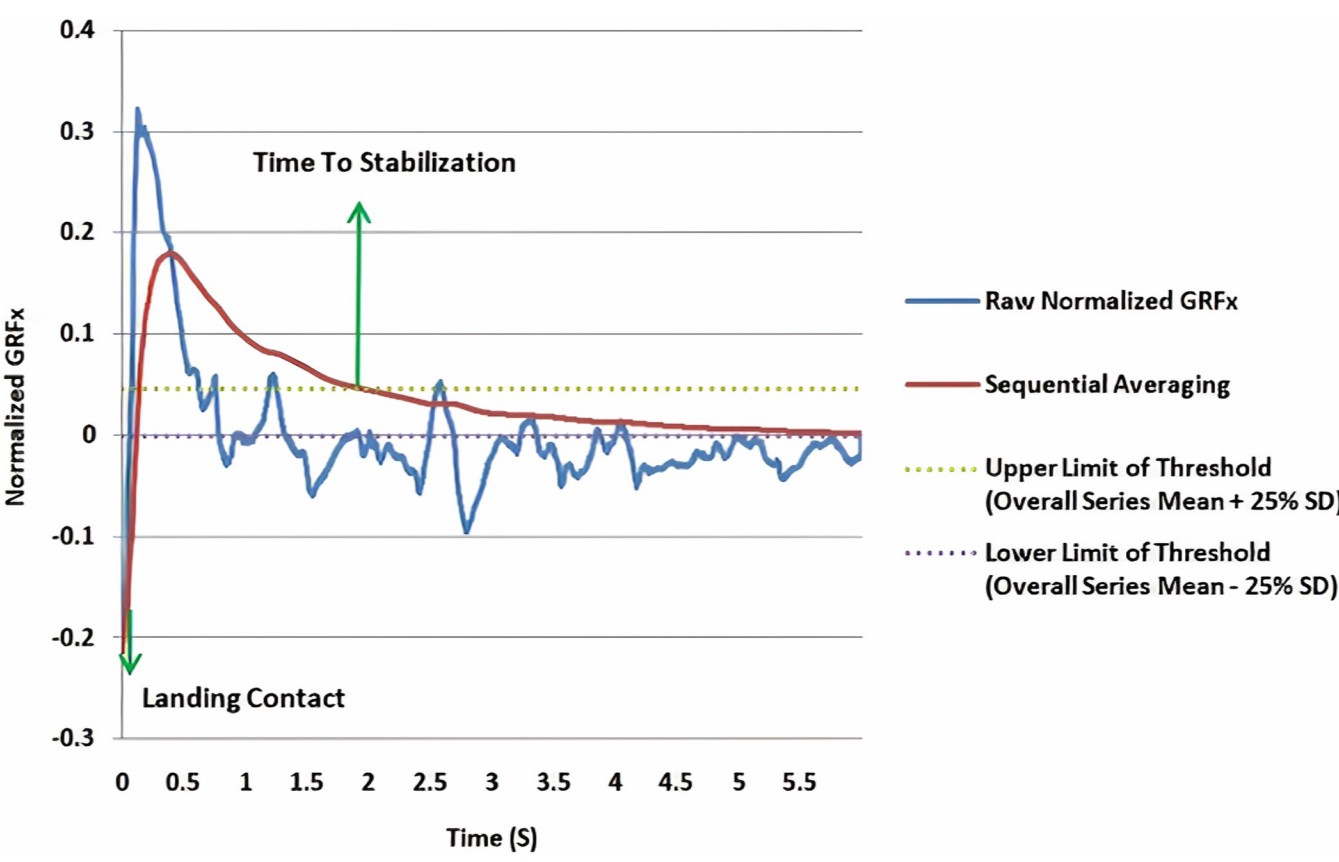

**Fig 2. Time to stabilization in the mediolateral direction.**

**2.5.3. Tampa Scale for Kinesiophobia (TSK).** The TSK was used to assess fear of movement. The questionnaire comprises eleven items, each with four response options: strongly disagree = 1, somewhat disagree = 2, somewhat agree = 3, and strongly agree = 4. The total score ranges from 11 to 44, with higher scores indicating greater fear of movement. The Persian version of TSK-11 has a acceptable internal consistency (Cronbach alpha 0.93) and excellent test-retest reliability (ICC = 0.93, 95% CI: 0.92–0.94) [33].

**2.5.4. Cumberland Ankle Instability Tool (CAIT).** The CAIT was used to measure the severity of perceived ankle instability. It comprises 9 questions, with a total possible score of 30, where higher scores indicate greater ankle stability. The Persian version of this questionnaire has good internal consistency (Cronbach's α of 0.78 for the right ankle and 0.79 for the left ankle) and substantial reliability ($ICC_{(2,\ 1)}$ = 0.88; 95% CI: 0.86–0.90) in athletes [34].

## 2.6. Groups

The groups consisted of exergaming and balance training. The treatment period for both groups included 12 sessions, occurring three times per week, with each session lasting 60 minutes.

**2.6.1. Exergaming.** Exergaming was conducted using a Wii Balance Board (Nintendo Co. Ltd., Kyoto, Japan), incorporating games such as Single Leg Extension, Torso Twist, Single Leg Twist, Sideways Leg Lift, Rowing Squat, Table Tilt, Penguin Fishing, Soccer Heading, Tightrope Walk, and Snowboard Slalom. The order of games in each session was based on the participants' preferences, with each game lasting 6 minutes to ensure inclusion of all games in

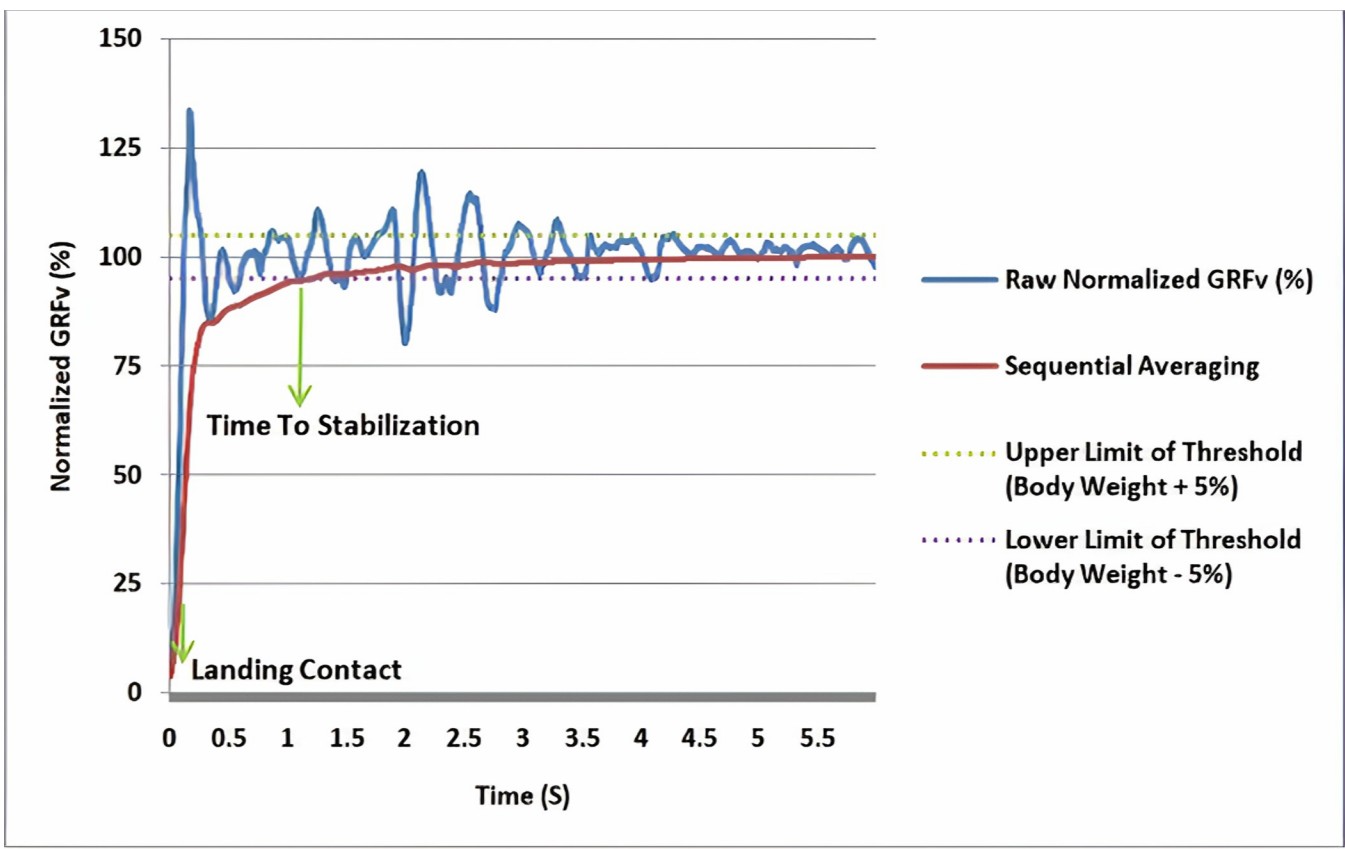

**Fig 3. Time to stabilization in the vertical direction.**

each session. Typically, each game offers three difficulty levels (beginner, advanced, and expert), with four sub-levels within each level (unstable, amateur, professional, and champion). Participants progressed to higher levels upon successfully completing their current level. Advancement through difficulty levels was managed by the Wii Fit game itself, with illuminated stars indicating participants' success in completing the game.

Participants received more stars and points for better balance. Some games required participants to maintain the COG within a designated yellow area and prevent it from falling out. In others, participants guided the game character to the desired destination by shifting weight in different directions at appropriate speeds and magnitudes. These games imposed cognitive demands on participants, requiring attention and rapid responses to visual stimuli. Additionally, games like Table Tilt and Penguin Slide required planning, while Soccer Heading necessitated decision-making and response inhibition [35, 36]. Therefore, the selected games appeared to encompass both cognitive and motor components.

These games, categorized as balance games in Wii Fit, are primarily intended to improve postural stability. Selected games, including the Single Leg Extension, Torso Twist, Single Leg Twist, Sideways Leg Lift, and Rowing Squat, require participants to maintain balance while executing various movements. This focus on dynamic balance is essential for athletes and individuals recovering from ankle injuries. The games simulate movements and challenges commonly faced in both sports and daily life. Additionally, weight-shifting activities in Snowboard Slalom, Table Tilt, Penguin Fishing, Soccer Heading, and Tightrope Walk closely reflect real-world scenarios, making them particularly relevant for athletes striving to regain dynamic

**Table 2. The description of games.**

| Games | Explains |
|---|---|
| Single Leg Extension | The participant balanced on one leg while swinging the opposite leg forward and backward and moving the upper limbs in opposite directions, ensuring that the COG stayed within a designated yellow circle shown on the screen. |
| Torso Twist | The participant sustained balance while twisting their upper body, ensuring that the COG remained within a specified yellow circle displayed on the screen. |
| Single Leg Twist | The participant sustained balance while lowering one hand down and raising the opposite knee up to touch it with the back of the hand, ensuring that the COG remained within a specified yellow circle shown on the screen. |
| Sideways Leg Lift | The participant maintained balance on one leg while lifting the other leg sideways and raising the opposite arm, ensuring that the COG remained within a designated yellow circle shown on the screen. |
| Rowing Squat | The participant balanced on their legs, pushing their hips back and bending their knees to about 120 degrees while simultaneously pulling their arms in towards their sides. The goal was to synchronize the squatting motion with the rowing movements. Visual cues displayed on the screen assisted players in maintaining proper form and timing, and they received feedback on their performance. This exercise targets various muscle groups, including the legs, core, and upper body, to improve strength, endurance, and coordination. |
| Table Tilt | The participant shifted their weight forward and side to side to align their COG with the base of support, guiding balls into corresponding holes. |
| Penguin Fishing | The participant shifted their weight forward and sideways to adjust the COG over their base of support, allowing them to control the movement of a penguin on ice and catch a fish. |
| Soccer Heading | The participant adjusted their COG to head the virtual soccer ball while striving to avoid colliding with other objects. |
| Tightrope Walk | The participant maintained rhythmic weight shifting to walk steadily without stumbling, while also navigating around obstacles. |
| Snowboard Slalom | The participant shifted their weight from side to side to guide the snowboarder through a series of gates as quickly as possible while maintaining balance and avoiding obstacles. |

COG = Center of Gravity

stability and confidence in their performance. Furthermore, many of these games, like Rowing Squat, engage multiple muscle groups, enhancing overall strength, coordination, and endurance. This comprehensive approach is crucial for rehabilitation, effectively preparing participants for the physical demands of their sports and daily activities. Further information regarding the games is provided in Table 2.

**2.6.2. Balance training.** The balance training included components such as single-limb stance, limb stance with ball kicking, single-limb hop to stabilization, and hop to stabilization and reach. More details about the training can be found in Table 3 [37–39]. Participants could advance to a higher level after successfully completing their current level. In other words, completing the task without errors was essential to move on to the next level of difficulty. The specific number of error-free repetitions required is detailed in Table 3. Throughout the training, the therapist offered precise verbal instructions to identify errors and reinforce correct actions. Over time, the instructions were gradually reduced to enable participants to successfully and independently perform the exercises.

## 2.7. Statistical analysis

For statistical analysis, IBM SPSS Statistics version 23.0 was utilized. The Shapiro-Wilk test was employed to assess the normality of the data distribution, which was found to be normally distributed except for age and the time since the last ankle sprain. An independent t-test was used to compare the two groups at baseline. For age and the time since the last ankle sprain,

Table 3. The description of balance training.

| | Difficulty Levels | Instructions | Criteria for Progression | Errors |
|---|---|---|---|---|
| **Single-Limb Stance** | 1) Eyes open, hard surface, 30 s, 3 Reps<br>2) Eyes open, hard surface, 60 s, 3 Reps<br>3) Eyes open, foam surface, 30 s, 3 Reps<br>4) Eyes open, foam surface, 60 s, 3 Reps<br>5) Eyes open, foam surface, 90 s, 3 Reps<br>6) Eyes open, foam surface, 30 s, ball toss, 3 Reps<br>7) Eyes open, foam surface, 60 s, ball toss, 3 Reps<br>8) Eyes open, foam surface, 90 s, ball toss, 3 Reps<br>9) Eyes closed, hard surface, 30 s, arms out, 3 Reps<br>10) Eyes closed, hard surface, 30 s, arms across, 3 Reps<br>11) Eyes closed, foam surface, 30 s, arms out, 3 Reps<br>12) Eyes closed, foam surface, 30 s, arms across, 3 Reps | The participant attempted to maintain balance on the affected limb while facing challenges such as altering the base of support, closing eyes, extending the duration, and throwing a ball [37, 38]. | If the participant could complete 3 repetitions without errors at each level of difficulty. | Errors were defined as:<br>A. Touching the ground with the opposite foot.<br>B. Excessive trunk motion (more than 30 degrees of lateral flexion).<br>C. Resting the opposite limb against the stance limb.<br>D. Lifting the hands from the chest while standing. |
| **Limb Stance with Ball Kicking** | 1) Double-limb stance, hard surface, ball Kicking, 3 Reps<br>2) Single- limb stance, hard surface, ball Kicking, 3 Reps<br>3) Double-limb stance, mini-trampoline, ball Kicking, 3 Reps<br>4) Single- limb stance, mini-trampoline, ball Kicking, 3 Reps | The participant was instructed to return the ball to the therapist and maintain balance as much as possible after kicking. During this training, the uninjured limb was used for kicking, while the affected limb served as the supporting limb [39]. | If the participant successfully completed 3 repetitions without any errors at each difficulty level. | Errors included A, B, C, and D mentioned earlier. |
| **Single-Limb Hop to Stabilization** | 1) Target at 18 inch, using arms to aid in stabilizing, 10 Reps<br>2) Target at 18 inch, hands on hips while stabilizing, 10 Reps<br>3) Target at 27 inch, using arms to aid in stabilizing, 10 Reps<br>4) Target at 27 inch, hands on hips while stabilizing, 10 Reps<br>5) Target at 36 inch, using arms to aid in stabilizing, 10 Reps<br>6) Target at 36 inch, hands on hips while stabilizing, 10 Reps | The participant was instructed to perform 10 hops in four directions:<br>1. Anterior-posterior<br>2. Medial-lateral<br>3. Anteromedial-posterolateral<br>4. Anterolateral-posteromedial<br>Three target distances (18, 27, or 36 inches) from the starting point were set. The participant hopped to these targets, stabilized their balance on one limb, then hopped back in the opposite direction to the starting position and stabilized again on one limb [37]. | If the participant successfully completed 10 repetitions without any errors at each level of difficulty. | Errors included A, B, C, and D mentioned earlier. |

(*Continued*)

**Table 3.** (Continued)

| | Difficulty Levels | Instructions | Criteria for Progression | Errors |
|---|---|---|---|---|
| **Hop to Stabilization and reach** | 1) Target at 18 inch, using arms to aid in stabilizing, 5 Reps 2) Target at 18 inch, hands on hips while stabilizing, 5 Reps 3) Target at 27 inch, using arms to aid in stabilizing, 5 Reps 4) Target at 27 inch, hands on hips while stabilizing, 5 Reps 5) Target at 36 inch, using arms to aid in stabilizing, 5 Reps 6) Target at 36 inch, hands on hips while stabilizing, 5 Reps | The participant hopped, stabilized, and reached back to the starting position. Then, they returned to the starting position and reached to the target position [37]. | If the participant successfully completed 5 repetitions without errors at each difficulty level. | Errors included A, B, C, and D mentioned earlier. |

Reps = Repetitions

which did not follow a normal distribution, the Mann-Whitney U test was utilized. In addition, the Chi-square test was used to compare the proportions of gender, affected limb, and dominant limb between the groups. The results indicated no significant differences between the groups, showing they were comparable at baseline. Although the gender distribution was not statistically significant between the groups, it was nearly significant ($p = 0.067$) and could potentially influence the results. Therefore, this variable was considered as a covariate to control for its effect. A two-way mixed ANCOVA was conducted with one within-subject factor (time: before, after, and follow-up) and one between-groups factor (group: exergaming and balance training) to examine the main effects and interaction between these factors. If a significant interaction effect was found, separate one-way repeated measures ANOVAs were conducted for each group to further investigate these interactions and assess the effect of time within each group individually. When the main effects were significant but the interaction was not, the main effects could be interpreted directly. Additionally, if a significant group effect was found, a one-way ANCOVA was conducted to identify which time points (post-treatment or follow-up) showed significant differences. Effect size was measured using partial eta squared ($\eta^2$), with interpretations as small (0.01–0.06), moderate (0.06–0.14), and large ($\geq 0.14$). When the time effect was significant, post hoc pairwise comparisons were conducted with Bonferroni corrections, and Cohen's d was used to evaluate the effect size (small = 0.2–0.5, medium = 0.5–0.8, large = 0.8–1.3, and very large $\geq 1.13$). It should be noted that the Intention to Treat (ITT) approach was used in the analysis. As shown in the CONSORT flowchart, out of the 34 participants, two from the intervention group and two from the control group did not participate in the evaluation one month post-treatment (follow-up). Missing data were addressed using the Expectation Maximization (EM) algorithm, which replaces missing values through the Maximum Likelihood (ML) approach. Thus, the analysis was conducted as if there were no dropouts, including data from all 34 participants.

## 3. Results

The descriptive data and the results of ANCOVA are presented in Tables 4 and 5, respectively.

**Table 4. The mean and standard deviation of parameters.**

| Parameters | Pre-Test | | | | Post-Test | | | | Follow-Up | | | |
|---|---|---|---|---|---|---|---|---|---|---|---|---|
| | Intervention | | Control | | Intervention | | Control | | Intervention | | Control | |
| | Mean | SD | Mean | SD | Mean | SD | Mean | SD | Mean | SD | Mean | SD |
| **Jump-Landing Test** | | | | | | | | | | | | |
| AP SI | 0.036 | 0.017 | 0.038 | 0.010 | 0.037 | 0.015 | 0.034 | 0.010 | 0.036 | 0.015 | 0.032 | 0.010 |
| ML SI | 0.143 | 0.027 | 0.144 | 0.022 | 0.136 | 0.024 | 0.155 | 0.017 | 0.122 | 0.019 | 0.135 | 0.015 |
| V SI | 0.203 | 0.057 | 0.211 | 0.031 | 0.197 | 0.050 | 0.202 | 0.038 | 0.188 | 0.036 | 0.192 | 0.047 |
| DPSI | 0.259 | 0.059 | 0.255 | 0.031 | 0.251 | 0.053 | 0.252 | 0.035 | 0.235 | 0.034 | 0.233 | 0.045 |
| AP TTS (ms) | 813.843 | 213.667 | 693.784 | 252.570 | 702.118 | 284.499 | 682.824 | 265.628 | 696.497 | 259.607 | 664.453 | 240.109 |
| ML TTS (ms) | 1891.765 | 40.316 | 1887.637 | 68.354 | 1864.118 | 64.116 | 1899.363 | 134.690 | 1884.218 | 70.588 | 1864.486 | 58.541 |
| V TTS (ms) | 1590.033 | 1034.597 | 1736.547 | 537.537 | 1600.647 | 757.081 | 1566.186 | 627.603 | 1471.138 | 664.284 | 1479.158 | 672.391 |
| RV TTS (ms) | 2085.120 | 107.072 | 2036.692 | 140.156 | 2018.852 | 159.223 | 2043.626 | 168.049 | 2032.619 | 141.571 | 1997.398 | 94.247 |
| **Clinical Tests** | | | | | | | | | | | | |
| Side-Hop Test (s) | 9.622 | 2.629 | 10.615 | 3.034 | 7.598 | 1.728 | 7.368 | 1.742 | 7.454 | 2.003 | 7.315 | 1.404 |
| TSK-11 (11–44) | 24.105 | 4.405 | 26.129 | 4.378 | 22.882 | 4.386 | 21.789 | 3.555 | 24.222 | 6.443 | 22.258 | 5.921 |
| CAIT (0–30) | 16.280 | 4.337 | 15.328 | 5.295 | 21.529 | 6.296 | 22.176 | 5.077 | 22.679 | 6.097 | 23.493 | 6.388 |

AP = Anteroposterior; ML = Mediolateral; V = Vertical; DPSI = Dynamic Postural Stability Index; SI = Stability Index; RV = Resultant Vector; TTS = Time to Stabilization; TSK = Tampa Scale for Kinesiophobia; CAIT = Cumberland Ankle Instability Tool; SD = Standard Deviation.

### 3.1. Stability Index (SI)

For the SI in the ML direction, the effects of time and group were statistically significant, while the interaction effect was not. The Bonferroni post hoc tests showed that in both groups, it decreased significantly one month after treatment compared to before and after treatment ($P = 0.013$ and $P<0.001$, respectively). Table 5 presents the Bonferroni results. Furthermore, the one-way ANCOVA test indicated a significant difference between the groups after treatment ($F(1,31) = 6.011$, $P = 0.020$, $\eta 2 = 0.162$) and one month after treatment ($F(1,31) = 4.889$, $P = 0.035$, $\eta 2 = 0.136$), with SI being significantly lower in the exergaming group than the balance training group at both time points.

For the DPSI, the time effect was statistically significant, whereas the effects of interaction and group were not. This indicates that both groups exhibited similar behavior over before, after, and follow-up, with neither group showing superiority over the other. The Bonferroni tests showed that, in both groups, the DPSI decreased significantly one month after treatment compared to both before and after treatment ($P = 0.040$ and $P = 0.018$, respectively).

### 3.2. Time to Stabilization (TTS)

The effects of time, group, and interaction did not show statistical significance.

### 3.3. Side-hop test

For performance assessed with the side-hop test, the time effect was statistically significant, whereas neither the interaction effect nor the group effect showed significance. This suggests that the performance change over time did not differ between the two groups, with neither group displaying superiority over the other. The Bonferroni tests revealed that in both groups, performance significantly improved (as observed by the decrease in the side-hop test) after treatment and one month after treatment compared to before treatment($P<0.001$).

**Table 5. The ANCOVA and Post_hoc results.**

| Parameters | Time Effect | | | Group Effect | | | Time* Group Effect | | | Post-hoc Tests |
|---|---|---|---|---|---|---|---|---|---|---|
| | F | P | η² | F | P | η² | F | P | η² | P, MD, 95% CI, Cohen's d |
| **Jump-Landing Test** | | | | | | | | | | |
| AP SI | 0.384 | 0.683 | 0.012 | 0.284 | 0.598 | 0.009 | 0.649 | 0.526 | 0.020 | |
| ML SI | 5.162 | **0.012*** | 0.143 | 4.253 | **0.048*** | 0.121 | 1.973 | 0.155 | 0.060 | **Follow-Up < Pre**, P = 0.013, MD: -0.015, 95% CI: -0.027 to -0.003, Cohen's d = 0.532 **Follow-Up < Post**, P<0.001, MD: -0.017, 95% CI: -0.025 to -0.008, Cohen's d = 0.859 |
| V SI | 1.392 | 0.256 | 0.043 | 0.231 | 0.634 | 0.007 | 0.037 | 0.937 | 0.001 | |
| DPSI | 3.225 | **0.047*** | 0.094 | 0.017 | 0.898 | 0.001 | 0.119 | 0.888 | 0.004 | **Follow-Up < Pre**, P = 0.040, MD:-0.023, 95% CI: -0.045 to -0.001, Cohen's d = 0.461 **Follow-Up < Post**, P = 0.018, MD: -0.017, 95% CI: -0.032 to -0.002, Cohen's d = 0.514 |
| AP TTS | 0.136 | 0.873 | 0.004 | 0.772 | 0.386 | 0.024 | 0.621 | 0.541 | 0.020 | |
| ML TTS | 0.886 | 0.418 | 0.028 | 0.032 | 0.858 | 0.001 | 2.548 | 0.086 | 0.076 | |
| V TTS | 1.249 | 0.288 | 0.039 | 0.042 | 0.839 | 0.001 | 0.326 | 0.662 | 0.010 | |
| RV TTS | 0.567 | 0.570 | 0.018 | 0.253 | 0.619 | 0.008 | 1.166 | 0.318 | 0.036 | |
| Side-Hop Test | 23.278 | **<0.001*** | 0.429 | 0.058 | 0.812 | 0.002 | 1.815 | 0.184 | 0.055 | **Post < Pre**, P<0.001, MD: -2.636, 95% CI: -3.526 to -1.746, Cohen's d = 1.250 **Follow-Up < Pre**, P<0.001, MD: -2.734, 95% CI: -3.834 to -1.635, Cohen's d = 1.078 |
| TSK-11 | 3.765 | **0.028*** | 0.105 | 0.082 | 0.777 | 0.003 | 2.057 | 0.136 | 0.060 | **Post < Pre**, P = 0.027, MD: -2.781, 95% CI: -5.306 to -0.256, Cohen's d = 0.467 |
| CAIT | 23.438 | **<0.001*** | 0.431 | 0.008 | 0.929 | 0.000 | 0.553 | 0.543 | 0.018 | **Post > Pre**, P<0.001, MD: 6.049, 95% CI: 3.609 to 8.488, Cohen's d = 1.086 **Follow-Up > Pre**, P<0.001, MD: 7.282, 95% CI: 4.718 to 9.845, Cohen's d = 1.255 |

AP = Anteroposterior; ML = Mediolateral; V = Vertical; DPSI = Dynamic Postural Stability Index; SI = Stability Index; RV = Resultant Vector; TTS = Time to Stabilization; TSK = Tampa Scale for Kinesiophobia; CAIT = Cumberland Ankle Instability Tool; MD = Mean Difference; CI = Confidence Interval.

η2 is effect size (small = 0.01–0.06, medium = 0.06–0.14 and large≥ 0.14). Cohen's d is effect size (small = 0.2–0.5, medium = 0.5–0.8, large = 0.8–1.3, and very large≥ 1.13).

* Significant differences (P < 0.05)

### 3.4. Tampa Scale for Kinesiophobia (TSK)

For the fear of movement assessed with the TSK, the time effect was statistically significant, while neither the interaction effect nor the group effect reached significance. Bonferroni tests demonstrated that in both groups, fear of movement significantly decreased (as evidenced by the decline in TSK scores) after treatment compared to before treatment (P = 0.027).

### 3.5. Cumberland Ankle Instability Tool (CAIT)

For the severity of perceived ankle instability evaluated using the CAIT, the effect of time was statistically significant, while neither the interaction effect nor the group effect showed significance. Bonferroni tests showed that in both groups, the scores of the CAIT questionnaire significantly increased after treatment and one month after treatment compared to before treatment. This indicates a decreasing trend in severity of ankle instability following the intervention (P<0.001).

## 4. Discussion

The results of the study indicated that both exergaming and balance training were effective in improving postural control during jump-landing. However, exergaming showed superior performance in the ML direction both after the treatment and one month later. In terms of

clinical and psychological outcomes, both groups demonstrated effectiveness, with neither group showing superiority over the other.

The SI and TTS were used to assess dynamic postural stability during jump-landing. No significant differences were observed in within-group and between-group comparisons for TTS. However, the DPSI and the SI in the ML direction showed significant differences within-group and between-group comparisons, respectively. This could be attributed to the characteristics of the SI, as previous studies have shown it has higher repeatability (ICC: 0.96) and greater accuracy (SEM: 0.03) than TTS [21]. Therefore, SI may be more capable of detecting small changes due to its greater stability and accuracy. Nevertheless, further research is required to investigate the sensitivity and specificity of these parameters to confirm this hypothesis.

In individuals with CAI, neuromuscular control during jump-landing is impaired [25]. Jump-landing is a complex dynamic task that requires the use of two postural control strategies to handle disturbances, such as the foot hitting the ground. These strategies include feed-forward (or open loop) strategies, which are proactively activated by the brain before the foot hits the ground to minimize the impact of the disturbance, and feedback (or closed loop) strategies, which are reactively triggered by mechanoreceptors after the foot hits the ground to compensate the effect of the disturbance and minimize potential damage [40]. The disruption in neuromuscular control often results from damage to mechanoreceptors caused by repeated ankle sprains. This damage impairs feedback-based strategies, resulting in an inadequate response to control the GRFs during foot contact with the ground [41]. Additionally, a chronic reduction in sensory messages from ankle structures to the spinal and supra-spinal centers leads to changes in the CNS and motor programs in the brain. Consequently, feed-forward strategies are impaired, reducing the body's readiness for movements like jump-landing [42]. This is evidenced by a delay and decrease in muscle activation, especially in the peroneal muscles [43, 44].

Furthermore, impairment of mechanoreceptors diminishes proprioception, affecting the capacity to perceive joint position, velocity, and movement direction. Consequently, this leads to increased errors during jump landings and raises the risk of re-injury [45].

Another factor contributing to compromised postural control during jump-landing may be muscle weakness, especially in the peroneal muscles. Research has identified arthrogenic inhibition as a prevalent cause of this weakness [46].

Based on the described mechanisms, it seems that improving mechanoreceptor function, minimizing muscle inhibition, and boosting muscle strength can result in enhanced proprioception and improved neuromuscular control of the ankle, ultimately affecting dynamic postural control during jump-landing. In a study carried out in 2024, a notable correlation was found between ankle muscle strength and joint proprioception with the TTS during jump-landing in healthy individuals. For those with CAI, there was a statistically significant correlation between the strength of the plantar flexors and the TTS in the AP direction. This suggests that the ankle muscles contribute to controlling the forward and backward movements of the body, preventing the COG from exceeding the posterior and anterior edges of the base of support, and reducing the impact of GRFs on the body [47].

The study found that postural control during jump-landing improved in both groups, particularly during exergaming. This improvement is probably due to the role of these groups in enhancing proprioception and neuromuscular control of the ankle. The balance training included various sensory-motor components; visual perception was manipulated by opening and closing the eyes, while proprioception was challenged by standing on hard and foam surfaces and reducing the base of support. Additionally, the exercises involved perturbations, requiring participants to manipulate a ball with the hands or feet in both stable and unstable positions. The aim of balance training was to strengthen the sensorimotor system through

continuous interaction between sensory input and motor output, as sensory information is crucial for selecting appropriate motor strategies and maintaining stability [48]. In conclusion, it appears that these exercises, incorporating both sensorimotor and perturbation-based components, effectively enhanced motor responses in dynamic and challenging environments by improving proprioceptive inputs. This improvement was reflected in better postural control during jump-landing.

In this regard, some studies have demonstrated that balance training can improve muscle activity patterns and reduce muscle activation time in individuals with CAI [49, 50]. Additionally, balance training effectively enhances the strength of the muscles surrounding the ankles [51]. Evidence also suggests that these exercises can improve proprioception in such individuals [52]. Therefore, we believe that balance training likely played a significant role in improving postural control during jump-landing by enhancing proprioception, increasing muscle strength, and restoring proper muscle activity patterns.

During exergaming, the brain continuously processes sensory inputs (proprioception, visual, and auditory) presented through the virtual environment of the game [9]. As previously mentioned, sensory inputs are crucial for neuromuscular and postural control. Exergaming offers a multi-component environment encompassing sensory, motor, and cognitive elements, likely enhancing proprioception, increasing muscle strength, restoring proper muscle activity patterns, and improving cognitive function, all of which contribute effectively to better postural control during jump-landing.

In this regard, Kim et al. (2015) and Kim et al. (2019) demonstrated that a 4-week program of video games using the Nintendo Wii Fit Plus device can significantly enhance proprioception and the strength of the plantar flexors, respectively, in individuals with CAI [53, 54]. However, no studies have yet been conducted to support the theory that improvements in postural control during jump-landing may result from restoring proper muscle activity patterns. Consequently, future research is strongly recommended to explore the impact of exergaming on recovering appropriate muscle activity patterns.

Cognition is another crucial factor significantly involved in exergaming, categorizing this intervention as a motor-cognitive intervention where various components of motor and cognitive functions are challenged [55]. In individuals with CAI, the speed of information processing in the brain decreases, leading to longer reaction times when responding to stimuli [15]. Studies have demonstrated that under dual-task conditions, where postural control is combined with complex cognitive tasks, performance significantly deteriorates because both tasks compete for limited brain resources [56]. On the other hand, jump- landing is a complex task that requires heightened integration of cognitive and motor functions, demanding significant attention and engagement of cognitive centers to execute it accurately. A 2023 meta-analysis revealed that dual-task training is more effective than single-task training or no intervention in improving balance (including performance on the Y balance test) in individuals with CAI [57]. Thus, the improvement in postural control during jump-landing after exergaming may be partly due to its role in reducing the cognitive load and making the movement more automatic. While some studies have shown that exergaming can improve static and dynamic balance (walking) in dual-task conditions among elderly individuals, no studies have been conducted on individuals with CAI. However, a 2021 study by Mohammadi et al. found that individuals with ankle instability performed better on simple and selective reaction time tests after 12 training sessions with the Nintendo Wii Fit Plus compared to a control group that received routine exercises. They suggested that video games can improve neurocognitive function in CAI [58]. It is important to note that in this study, the jump-landing task was performed under single-task conditions, without a simultaneous cognitive task, which would have made it more challenging. Consequently, the control group's success in improving postural

control may be attributed to the balance exercises' emphasis on sensory and motor components, which were sufficient for enhancing postural control in jump-landing under single-task conditions. However, if the test was conducted under dual-task conditions, the difference between the exergaming and traditional balance training might have been more pronounced. Therefore, it is strongly recommended that future studies incorporate the jump-landing under dual-task conditions along with other assessments.

The study results indicated that exergaming significantly improved stability in the ML direction compared to balance training, with these effects lasting up to one month post-treatment. This enhancement may be attributed to the emphasis on ML weight shifting in various games, such as single-leg swings, leg raises, table tilts, penguin fishing, soccer heading, tightrope walking, and snowboard slalom. Repeated practice of these ML movements, which demand precise and coordinated control, fosters motor learning. Motor learning involves changes in the nervous system that improve the efficiency and accuracy of specific movements. This process is essential for enhancing postural stability and balance, as it aids individuals in maintaining and controlling posture, adapting to disturbances, and integrating sensory and motor information effectively [59]. Training programs that emphasize motor learning can significantly improve postural stability and overall balance. Consequently, the enhanced stability in the ML direction can be attributed to the role of exergaming in facilitating motor learning and, therefore, improving postural stability. Supporting these findings, a 2019 study by Kim et al. demonstrated that for individuals with CAI, video game training was more effective than traditional exercises in improving dynamic balance (levels 2, 4, and 8) on the Biodex system in the ML direction [18]. Although the balance measurement tools differed between the studies (standing on Biodex vs. jump-landing), the games used were quite similar, except for Penguin Fishing in Kim's study. Therefore, the similar outcomes can be attributed to the use of comparable games.

The finding that stability in the ML direction was significantly greater with exergaming compared to balance training, even up to one month post-treatment, underscores the long-term benefits of exergaming and its strength as a treatment. These results may be linked to exercise-dependent neuroplasticity. Video games, through repetitive practice and constant sensory feedback, serve as powerful tools for long-term reinforcement and motor learning. Studies involving elderly individuals or those with neurological disorder such as stroke, using brain imaging tools like electroencephalography and functional magnetic resonance imaging (FMRI), have shown that video games can influence brain activity [9, 60]. The findings suggest that following exergaming, the brain becomes more efficient and requires fewer neural resources for motor preparation, reflecting training- induced neural plasticity [9]. One reason for this is that video games rely on movement observation and visual-motor processing, which activate mirror neurons in the brain. Mirror neurons play a crucial role in reorganizing the cerebral cortex [9]. Prochnow et al. using FMRI on healthy subjects demonstrated that areas of the brain associated with mirror neurons are specifically activated during exergaming [61]. Therefore, video games appear to have lasting effects that extend beyond the treatment period.

In Kim's study, within-group comparisons showed that both the exergaming using Nintendo Wii Fit Plus and the traditional exercises, which included strengthening and balance training, had significant improvements in dynamic balance on the Biodex system after 12 sessions. Despite the shorter intervention duration in this study compared to ours (20 minutes per session versus 60 minutes), the results demonstrated that both groups effectively improved dynamic balance in individuals with CAI [18].

In a 2021 study by Shousha et al. participants with CAI were divided into three groups: balance training, balance training plus Biodex exercises, and balance training plus virtual reality exercises. Over three months, all groups significantly improved dynamic balance (level 8) on

the Biodex system. However, the group that combined balance training with virtual reality and Biodex exercises showed more impressive results compared to the balance training-only group. The control group in Shousha's study, similar to the control group in our study, included exercises such as single-limb stance, single-limb hop to stabilization, and hop to stabilization and reach. Despite the different balance evaluation methods (standing on Biodex system versus jump-landing), both studies demonstrated that balance training effectively improves dynamic balance [19].

In this study, it was found that both groups had a positive effect on clinical and psychological measures. Specifically, the performance, as assessed by the side-hop test, and the severity of perceived ankle instability, as measured by the CAIT, demonstrated significant improvement in both groups after the treatment and also one month post-treatment compared to before. Additionally, the fear of movement, evaluated using the TSK, was significantly reduced in both groups after the treatment compared to before. The sustained significant improvement in clinical measures one month after the treatment compared to before is a noteworthy clinical finding, indicating that even after one month, the performance and perceived ankle stability remained favorable compared to before. Regarding the fear of movement, which is a psychological measure, the results indicate that both groups effectively reduced it, but there is a possibility of its recurrence after one month post-treatment.

Reducing perceived ankle instability and alleviating the fear of movement are crucial factors in enhancing self-confidence. Hence, a plausible explanation for the improvement in postural control during jump-landing in both groups could be attributed to enhanced perceived ankle stability, a reduction in fear and anxiety, and consequently, an increase in self-assurance.

The enhanced perceived ankle stability following treatments might result from the correction of behavioral-motor dysfunctions in these individuals. Indeed, improvements in muscle activity timing, reduction in neuromuscular inhibition, increased muscle strength, and enhancement of movement patterns can manifest as increased joint stability [62]. Moreover, the increased joint stability could be associated with the improved joint proprioception. Proprioception plays a crucial role, and even minor inaccuracies in it could cause inappropriate foot contact with the ground during sports activities, leading to a sense of instability [63]. Consequently, by enhancing proprioception and the individual's awareness of joint position in space, a greater perceived joint stability may be achieved. In line with the results of present study, a study by Kim et al. in 2015 found that participating in exergaming using Nintendo Wii Fit Plus for 12 sessions of 20 minutes significantly reduced the perceived ankle instability [53]. Furthermore, a meta-analysis conducted in 2024 showed that traditional balance training, when compared to a control group without treatment, as well as strengthening exercises, could decrease the ankle instability [7].

Regarding the effectiveness of exergaming on reducing the fear of movement, this study is the first investigation conducted in individuals with CAI. A meta-analysis conducted in 2022 on individuals with chronic musculoskeletal pain (including back pain, neck pain, sciatica, osteoarthritis, and fibromyalgia) found that video games were more effective in reducing fear of movement compared to other treatments [64]. Additionally, another meta-analysis in 2021 revealed that virtual reality was effective in improving fear of movement at post-treatment and 6-month follow-up in people with low back pain [65].

Regarding the performance improvement observed in both groups, the findings of this study align with those of Mohammadi et al in 2021. In that study, the performance of individuals with CAI was assessed through hop tests. The study suggested that both groups (routine exercises and video games using Nintendo Wii Fit Plus) significantly contribute to performance enhancement, with no superiority of one group over the other [58].

## 5. Limitations

Recording muscle activity through electromyography alongside kinetic parameters could aid in interpreting the results. Furthermore, conducting the jump-landing test under both single and dual-task conditions could elucidate the cognitive component's role in exergaming.

## 6. Conclusion

The results of the study indicated that both exergaming and balance training were effective in improving postural control during jump-landing. However, exergaming showed superior performance in the ML direction both after the treatment and one month later. In terms of clinical and psychological outcomes, both groups demonstrated effectiveness, with neither group showing superiority over the other.

## Supporting information

**S1 Checklist. CONSORT 2010 checklist of information to include when reporting a randomized trial.**
(DOC)

**S1 Text. Study protocol (English version).**
(DOCX)

**S2 Text. Study protocol (Persian version).**
(DOCX)

**S1 Dataset. Participant's data including characteristics and primary outcome measurement.**
(XLSX)

## Author Contributions

**Conceptualization:** Sadaf Sepasgozar Sarkhosh, Roya Khanmohammadi.

**Data curation:** Sadaf Sepasgozar Sarkhosh.

**Formal analysis:** Sadaf Sepasgozar Sarkhosh, Roya Khanmohammadi.

**Funding acquisition:** Roya Khanmohammadi.

**Investigation:** Zeinab Shiravi.

**Project administration:** Roya Khanmohammadi.

**Supervision:** Roya Khanmohammadi.

**Writing – original draft:** Sadaf Sepasgozar Sarkhosh.

**Writing – review & editing:** Roya Khanmohammadi, Zeinab Shiravi.

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
