## [Decision Letter · Decision Letter 0]

8 Oct 2024

PONE-D-24-32029Comparison of the effects of exergaming and balance training on dynamic postural stability during jump-landing in recreational athletes with chronic ankle instabilityPLOS ONE

Dear Dr. Khanmohammadi,

Thank you for submitting your manuscript to PLOS ONE. After careful consideration, we feel that it has merit but does not fully meet PLOS ONE’s publication criteria as it currently stands. Therefore, we invite you to submit a revised version of the manuscript that addresses the points raised during the review process.

We look forward to receiving your revised manuscript.

Kind regards,

Žiga Kozinc

Academic Editor

PLOS ONE

Journal Requirements:

“This project was funded by the Tehran University of Medical Sciences (Grant No. 1401-2-103-66959).”

3. In this instance it seems there may be acceptable restrictions in place that prevent the public sharing of your minimal data. However, in line with our goal of ensuring long-term data availability to all interested researchers, PLOS’ Data Policy states that authors cannot be the sole named individuals responsible for ensuring data access (http://journals.plos.org/plosone/s/data-availability#loc-acceptable-data-sharing-methods).

Reviewers' comments:

Reviewer's Responses to Questions

**Comments to the Author**

1. Is the manuscript technically sound, and do the data support the conclusions?

Reviewer #1: Partly

Reviewer #2: Yes

Reviewer #3: Yes

2. Has the statistical analysis been performed appropriately and rigorously? 

Reviewer #1: No

Reviewer #2: Yes

Reviewer #3: Yes

3. Have the authors made all data underlying the findings in their manuscript fully available?

Reviewer #1: Yes

Reviewer #2: Yes

Reviewer #3: Yes

4. Is the manuscript presented in an intelligible fashion and written in standard English?

Reviewer #1: Yes

Reviewer #2: Yes

Reviewer #3: Yes

5. Review Comments to the Author

Reviewer #1: A two-arm randomized controlled clinical trial was conducted which aimed to determine if exergaming is more effective than balance training in improving dynamic postural control during jump-landing movements among athletes with chronic ankle instability. Secondary aims included comparing the effectiveness of intervention on clinical and psychological outcomes. The results are unclear due to the lack of details about how the interaction and main effects were tested.

Minor revisions:

1- Line 192: State the statistical testing method which achieves 80% power.

2- Line 322: For improved clarity, replace frequency with proportions.

3- Table 1: In addition to the frequencies, state the corresponding percentages of female, left affected side, and left dominant side.

4- Clarify if interaction effects were tested according to standard practice.

If the interaction effect is significant, provide an interpretation of the results, but do not test main effects because the tests for main effects are uninteresting in light of significant interactions. If interaction effects are non-significant, drop the interaction effects from the model and test the main effects. Determining which results to present when testing interactions is often a multi-step process.

5- Cite the statistical software used for the analysis.

Reviewer #2: The authors responded to questions accurately. The results of manuscript are helpful in the designing the therapeutic exercises for improving balance control in recreational athletes with CAI. The manuscript is appropriate for publication in this journal.

Reviewer #3: Dear Authors,

Kindly address the following:

1. Participants inclusion and exclusion criteria should be more clear.

2. Primary outcomes measures and Secondary outcome measures, kindly provide more reason in detail about their selection in this study.

3. A force plate device (Bertec Corporation, Columbus, OH, USA)- Reliability and validity of this device is needed.

4. The description of games- Can you able to provide the reasons foe selecting these specific games.

5. Discussion was well written and covered all the aspects.

6. English language correction is needed in some parts of this study.

6. PLOS authors have the option to publish the peer review history of their article (what does this mean?). If published, this will include your full peer review and any attached files.

Reviewer #1: No

Reviewer #2: No

Reviewer #3: **Yes: **KAKARAPARTHI V NAGARAJ

---

## [Author Response · Author response to Decision Letter 0]

11 Oct 2024

Dear Reviewer 1

We sincerely appreciate your thorough review of the manuscript and the valuable comments and suggestions provided. We believe that the manuscript has significantly improved as a result of incorporating your feedback, which has been highlighted in red throughout the text.

Question 1: A two-arm randomized controlled clinical trial was conducted which aimed to determine if exergaming is more effective than balance training in improving dynamic postural control during jump-landing movements among athletes with chronic ankle instability. Secondary aims included comparing the effectiveness of intervention on clinical and psychological outcomes. The results are unclear due to the lack of details about how the interaction and main effects were tested. Clarify if interaction effects were tested according to standard practice.

If the interaction effect is significant, provide an interpretation of the results, but do not test main effects because the tests for main effects are uninteresting in light of significant interactions. If interaction effects are non-significant, drop the interaction effects from the model and test the main effects. Determining which results to present when testing interactions is often a multi-step process.

Answer 1: We value your feedback; however, it’s important to highlight that this aspect was also addressed in our statistical analysis. To provide further clarity on this matter, additional explanations have been included in the statistical analysis section. 

“If a significant interaction effect was found, separate one-way repeated measures ANOVAs were conducted for each group to further investigate these interactions and assess the effect of time within each group, individually. When the main effects were significant but the interaction was not, the main effects could be interpreted directly”. 

Furthermore, as shown in the results, the interaction effect is not significant for any of the variables, allowing for a straightforward interpretation of the main effects.

Question 2: Line 192: State the statistical testing method which achieves 80% power.

Answer 2: In response to your feedback, we have revised the "Sample Size" section as follows.

“The sample size was estimated from a pilot study, focusing on SI and TTS in the ML direction. An a priori power analysis was conducted using G*Power 3.1.9.2 software to achieve 80% power. The statistical testing method chosen was a repeated measures ANOVA with a within-between interaction design. The calculation used a effect size of 0.054 and 0.061, a power of 0.8, α = 0.05, non-sphericity correction (ε) = 1, and a correlation among repeated measures of 0.5. The analysis indicated that at least 30 participants were needed to detect a within-between interaction effect in a test design of 2 groups and 3 measurements with the specified parameters. Consequently, 34 participants were recruited to account for a potential 10% dropout rate.”

Question 3: Line 322: For improved clarity, replace frequency with proportions.

Answer 3: We replaced frequency with proportions.

Question 4: Table 1: In addition to the frequencies, state the corresponding percentages of female, left affected side, and left dominant side.

Answer 4: We included the percentages in the Table 1.

Question 5: Cite the statistical software used for the analysis.

Answer 5: We added the following sentence to the text.

“For statistical analysis, IBM SPSS Statistics version 23.0 was utilized.”

Dear Reviewer 3

We sincerely appreciate your thorough review of the manuscript and the valuable comments and suggestions provided. We believe that the manuscript has significantly improved as a result of incorporating your feedback, which has been highlighted in blue throughout the text.

Question 1: Participants inclusion and exclusion criteria should be clearer.

Answer 1: We have revised this section as follows.

 “Inclusion Criteria:

1. Age between 18 and 40 years.

2. A self-reported history of at least one significant acute ankle sprain that:

o Occurred more than 12 months prior to study enrollment.

o Was associated with inflammatory symptoms, such as pain and swelling.

o Resulted in at least one day of interrupted physical activity [2].

3. The most recent ankle sprain occurred more than 3 months before study enrollment [2].

4. A history of at least two episodes of "giving way," recurrent ankle sprains, or "feelings of ankle joint instability" within the last 6 months:

o "Giving way" refers to uncontrolled and unpredictable episodes of excessive inversion of the rear foot during activities, which do not result in an acute sprain.

o "Recurrent sprains" are defined as two or more sprains to the same ankle.

o "Feelings of instability" refer to the perception of instability during daily activities or sports, often associated with the fear of a new ankle sprain [2].

5. A Cumberland Ankle Instability Tool (CAIT) score of ≤ 24 (the CAIT is scored from 0–30, with lower scores indicating more instability) [2].

6. Participants must be recreationally active, engaging in sports that involve jumping (such as volleyball, basketball, soccer, or handball) at least three times a week, for a minimum of 30 minutes per session [27].

7. No self-reported history of fractures or surgery in the lower extremity.

8. No known psychological or neurological disorders.

9. No prior experience with video game-based training.

Exclusion Criteria:

1. If participants missed three or more non-consecutive training sessions, or two consecutive sessions.

2. They unable to perform the required maneuvers. »

Question 2: Primary outcomes measures and Secondary outcome measures, kindly provide more reason in detail about their selection in this study.

Answer 2: We revised this section to “The primary outcome measures included the SI and the TTS in the anteroposterior (AP), mediolateral (ML), and vertical (V) directions, along with the dynamic postural stability index (DPSI) and the resultant vector time to stabilization (RVTTS) during jump landing. The SI and TTS were selected for their strong reliability as established metrics for evaluating dynamic balance during jump landings. These parameters can distinguish individuals with CAI from those who are healthy, as research show that individuals with CAI typically achieve higher scores on the SI and TTS than their healthy counterparts [25]. Furthermore, these parameters are associated with an increased risk of recurrent injuries [26].

These measures enable the evaluation of neuromuscular control, which is crucial in minimizing the risk of re-injury and improving functional outcomes in sports. Thus, improving these parameters and ensuring sufficient stability during jump landings is a vital aspect of the rehabilitation program.

Secondary outcomes included performance metrics, fear of movement, and the severity of perceived ankle instability, assessed using the side-hop test, the Tampa Scale for Kinesiophobia (TSK), and the Cumberland Ankle Instability Tool (CAIT), respectively. These secondary outcomes were selected to provide a more holistic evaluation of the participants' functional and psychological well-being.

The side-hop test measures agility and functional performance of the ankle during dynamic activities, while the TSK evaluates fear of movement or re-injury—an important factor that can significantly impact rehabilitation and athletic performance. The CAIT assesses participants' subjective perceptions of ankle instability, which is essential for understanding their overall ankle function. Incorporating these secondary measures enhances the primary dynamic postural stability outcomes by considering both functional performance and psychological factors that play a crucial role in the rehabilitation process and long-term recovery for athletes with CAI.”

Question 3: A force plate device (Bertec Corporation, Columbus, OH, USA)- Reliability and validity of this device is needed.

Answer 3: We have incorporated the following sentences to address your comment. 

“The Bertec force plate is well-known for its exceptional reliability and validity in assessing ground reaction forces (GRF). Numerous studies have shown that it provides accurate and consistent data across different movement patterns, establishing it as a "gold standard" for evaluating balance and center of pressure (COP) measurements [28-30].”

Question 4: The description of games- Can you able to provide the reasons for selecting these specific games.

Answer 4: We added the following paragraph to the text 

“These games, categorized as balance games in Wii Fit, are primarily intended to improve postural stability. Selected games, including the Single Leg Extension, Torso Twist, Single Leg Twist, Sideways Leg Lift, and Rowing Squat, require participants to maintain balance while executing various movements. This focus on dynamic balance is essential for athletes and individuals recovering from ankle injuries. The games simulate movements and challenges commonly faced in both sports and daily life. Additionally, weight-shifting activities in Snowboard Slalom, Table Tilt, Penguin Fishing, Soccer Heading, and Tightrope Walk closely reflect real-world scenarios, making them particularly relevant for athletes striving to regain dynamic stability and confidence in their performance. Furthermore, many of these games, like Rowing Squat, engage multiple muscle groups, enhancing overall strength, coordination, and endurance. This comprehensive approach is crucial for rehabilitation, effectively preparing participants for the physical demands of their sports and daily activities.”

Question 5: English language correction is needed in some parts of this study.

Answer 5: The manuscript was reviewed by a native English speaker.

---

## [Decision Letter · Decision Letter 1]

5 Nov 2024

PONE-D-24-32029R1Comparison of the effects of exergaming and balance training on dynamic postural stability during jump-landing in recreational athletes with chronic ankle instabilityPLOS ONE

Dear Dr. Khanmohammadi,

Thank you for submitting your manuscript to PLOS ONE. After careful consideration, we feel that it has merit but does not fully meet PLOS ONE’s publication criteria as it currently stands. Therefore, we invite you to submit a revised version of the manuscript that addresses the points raised during the review process.

We look forward to receiving your revised manuscript.

Kind regards,

Žiga Kozinc

Academic Editor

PLOS ONE

**Journal Requirements:**

Reviewers' comments:

Reviewer's Responses to Questions

**Comments to the Author**

1. If the authors have adequately addressed your comments raised in a previous round of review and you feel that this manuscript is now acceptable for publication, you may indicate that here to bypass the “Comments to the Author” section, enter your conflict of interest statement in the “Confidential to Editor” section, and submit your "Accept" recommendation.

Reviewer #1: (No Response)

2. Is the manuscript technically sound, and do the data support the conclusions?

Reviewer #1: Yes

3. Has the statistical analysis been performed appropriately and rigorously? 

Reviewer #1: Yes

4. Have the authors made all data underlying the findings in their manuscript fully available?

Reviewer #1: Yes

5. Is the manuscript presented in an intelligible fashion and written in standard English?

Reviewer #1: Yes

6. Review Comments to the Author

**Reviewer #1:** Minor revisions:

1- Table 1: Check the data for normal distributions. If variables do not follow normal distributions, summarize using median, first and third quartiles and compare using Wilcoxon rank sum test.

7. PLOS authors have the option to publish the peer review history of their article (what does this mean?). If published, this will include your full peer review and any attached files.

Reviewer #1: No

---

## [Author Response · Author response to Decision Letter 1]

6 Nov 2024

Dear Reviewer 1

We sincerely appreciate your thorough review of the manuscript and the valuable comments and suggestions provided. We believe that the manuscript has significantly improved as a result of incorporating your feedback, which has been highlighted in red throughout the text.

Question 1: 1- Table 1: Check the data for normal distributions. If variables do not follow normal distributions, summarize using median, first and third quartiles and compare using Wilcoxon rank sum test 

Answer 1: Thank you for your attention to this detail. As previously noted, all variables except for age and time since the last ankle sprain followed a normal distribution. Consequently, we revised Table 1 to present the median and interquartile range (Q1 and Q3) for these two variables in place of the mean and standard deviation.

---

## [Decision Letter · Decision Letter 2]

15 Nov 2024

Comparison of the effects of exergaming and balance training on dynamic postural stability during jump-landing in recreational athletes with chronic ankle instability

PONE-D-24-32029R2

Dear Dr. Khanmohammadi,

We’re pleased to inform you that your manuscript has been judged scientifically suitable for publication and will be formally accepted for publication once it meets all outstanding technical requirements.

Kind regards,

Žiga Kozinc

Academic Editor

PLOS ONE

Additional Editor Comments (optional):

Reviewers' comments:

Reviewer's Responses to Questions

**Comments to the Author**

1. If the authors have adequately addressed your comments raised in a previous round of review and you feel that this manuscript is now acceptable for publication, you may indicate that here to bypass the “Comments to the Author” section, enter your conflict of interest statement in the “Confidential to Editor” section, and submit your "Accept" recommendation.

Reviewer #1: All comments have been addressed

2. Is the manuscript technically sound, and do the data support the conclusions?

Reviewer #1: (No Response)

3. Has the statistical analysis been performed appropriately and rigorously? 

Reviewer #1: (No Response)

4. Have the authors made all data underlying the findings in their manuscript fully available?

Reviewer #1: (No Response)

5. Is the manuscript presented in an intelligible fashion and written in standard English?

Reviewer #1: (No Response)

6. Review Comments to the Author

Reviewer #1: (No Response)

7. PLOS authors have the option to publish the peer review history of their article (what does this mean?). If published, this will include your full peer review and any attached files.

Reviewer #1: No

---

## [Editor Report · Acceptance letter]

4 Dec 2024

PONE-D-24-32029R2 

PLOS ONE

Dear Dr. Khanmohammadi, 

I'm pleased to inform you that your manuscript has been deemed suitable for publication in PLOS ONE. Congratulations! Your manuscript is now being handed over to our production team.

Kind regards, 

on behalf of

Dr. Žiga Kozinc 

Academic Editor

PLOS ONE